# Increasing Fruit and Vegetable Variety over Time Is Associated with Lower 15-Year Healthcare Costs: Results from the Australian Longitudinal Study on Women’s Health

**DOI:** 10.3390/nu13082829

**Published:** 2021-08-18

**Authors:** Jennifer N. Baldwin, Lee M. Ashton, Peta M. Forder, Rebecca L. Haslam, Alexis J. Hure, Deborah J. Loxton, Amanda J. Patterson, Clare E. Collins

**Affiliations:** 1Priority Research Centre for Physical Activity and Nutrition, University of Newcastle, Callaghan, NSW 2308, Australia; jennifer.baldwin@newcastle.edu.au (J.N.B.); lee.ashton@newcastle.edu.au (L.M.A.); rebecca.williams@newcastle.edu.au (R.L.H.); amanda.patterson@newcastle.edu.au (A.J.P.); 2School of Health Sciences, College of Health, Medicine and Wellbeing, University of Newcastle, Callaghan, NSW 2308, Australia; 3Priority Research Centre for Generational Health and Ageing, University of Newcastle, Callaghan, NSW 2308, Australia; peta.forder@newcastle.edu.au (P.M.F.); alexis.hure@newcastle.edu.au (A.J.H.); deborah.loxton@newcastle.edu.au (D.J.L.); 4School of Medicine and Public Health, College of Health, Medicine and Wellbeing, University of Newcastle, Callaghan, NSW 2308, Australia; 5Hunter Medical Research Institute, New Lambton Heights, NSW 2305, Australia

**Keywords:** fruit, vegetables, diet quality, health care costs, Medicare, women’s health

## Abstract

Healthcare costs are lower for adults who consume more vegetables; however, the association between healthcare costs and fruit and vegetable varieties is unclear. Our aim was to investigate the association between (i) baseline fruit and vegetable (F&V) varieties, and (ii) changes in F&V varieties over time with 15-year healthcare costs in an Australian Longitudinal Study on Women’s Health. The data for Survey 3 (*n* = 8833 women, aged 50–55 years) and Survey 7 (*n* = 6955, aged 62–67 years) of the 1946–1951 cohort were used. The F&V variety was assessed using the Fruit and Vegetable Variety (FAVVA) index calculated from the Cancer Council of Victoria’s Dietary Questionnaire for Epidemiological Studies food frequency questionnaire. The baseline FAVVA and change in FAVVA were analysed as continuous predictors of Medicare claims/costs by using multiple regression analyses. Healthy weight women made, on average, 4.3 (95% confidence interval (CI) 1.7–6.8) fewer claims for every 10-point-higher FAVVA. Healthy weight women with higher fruit varieties incurred fewer charges; however, this was reversed for women overweight/obese. Across the sample, for every 10-point increase in FAVVA over time, women made 4.3 (95% CI 1.9–6.8) fewer claims and incurred $309.1 (95% CI $129.3–488.8) less in charges over 15 years. A higher F&V variety is associated with a small reduction in healthcare claims for healthy weight women only. An increasing F&V variety over time is associated with lower healthcare costs.

## 1. Introduction

An adequate intake of fruit and vegetables (F&V) is associated with a lower risk of cardiovascular disease, cancer and all-cause mortality [1,2]. However, globally, 78% of adults consume less than five daily portions of F&V [3]. Low F&V consumption is among the leading dietary risk factors for mortality, each accounting for more than 2% of global deaths [4].

Poor diets are costly for governments. In Canada, inadequate F&V intake generates an economic burden of approximately $CAN 3.3 billion per year, including $CAN 1 billion in healthcare costs [5]. The results from two previous cohort studies have shown that the cumulative healthcare costs were lower for men [6] and women [7] who consumed more vegetables. However, the evidence for the association with fruit intake is mixed. The Chicago Western Electric Study reported lower healthcare costs for men in the highest tertile of fruit intake [6], while our analysis of Australian women found that healthcare costs increased as fruit consumption increased [7].

The variety of F&V intake, in addition to frequency, is important for health benefits. The dietary guidelines in the US, UK and Australia all recommend increasing the F&V variety to increase both the quantity and diversity of the nutrients consumed [8,9,10]. The frequency and variety of the F&V intake were each inversely associated with the risk of type 2 diabetes [11]. Green leafy vegetables were associated with a reduced risk of coronary heart disease (relative risk (RR) 0.83 (95% CI 0.75–0.91)), while cruciferous vegetables were associated with a reduced risk of cancer (RR 0.84 (95% CI 0.72–0.97)) [1].

Knowledge of the association between the F&V variety with healthcare costs is lacking and could potentially inform future approaches to population dietary interventions and health policies. The aim of the current study of mid-aged women in the Australian Longitudinal Study on Women’s Health (ALSWH) was to investigate the association between (i) Part I: the F&V variety at the baseline and (ii) Part II: changes in the F&V variety over time with cumulative 15-year healthcare costs.

## 2. Materials and Methods

### 2.1. Australian Longitudinal Study on Women’s Health (ALSWH)

This study uses data from the ALSWH [12]. Women in three cohorts (born 1973–1978, 1946–1951 and 1921–1926) were randomly selected from the Medicare database (Australia’s government funds universal health coverage, which includes all permanent residents) to take part in Survey 1 in 1996 [12]. The surveys were initially mailed; however, participants have had the option of online surveys since 2011. Participants living in rural/remote areas were intentionally oversampled [13]. The original sample was a representative sample of over 40,000 Australian women, although women from non-English speaking backgrounds were under-represented [14]. Ethical approvals were granted by the University of Newcastle (h-076-0795) and the University of Queensland (200400224). Medicare data consent was provided for the overall ALSWH study, and the use of the linked Medicare Benefits Schedule data was granted by the ALSWH Data Access Committee.

### 2.2. Participants: The 1946–1951 Cohort

For this analysis, the data for Survey 3 (2001) (*n* = 11,228 women, aged 50–55 years) and Survey 7 (2013) (*n* = 9151, then aged 62–67 years) of the 1946–1951 cohort were used. The response rates for Surveys 3 and 7 were 85% and 81%, respectively, excluding women who had died or withdrawn since Survey 1 [15].

One thousand and seven (*n* = 1007, 7.3%) women opted out of the Medicare Benefits Schedule (MBS) data linkage. Women with the highest 1% (*n* = 128, total charges > $79,525) and lowest 1% (*n* = 445, total charges = $0) of cumulative MBS charges over the 15-year period were excluded from the analyses to avoid anomalies associated with extreme values. The Fruit and Vegetable Variety Index (FAVVA) data were available for *n* = 9526 during Survey 3 and *n* = 7648 during both Survey 3 and Survey 7. Women with missing/incomplete dietary data were more likely to find it difficult/impossible to manage on their current income, live in outer regional Australia, be a smoker and be sedentary and were less likely to have a healthy body mass index (BMI) compared with women who had data at both time points. Baseline BMI data were missing for *n* = 569 women, residential area data for *n* = 37 women and socioeconomic data for *n* = 87 women, who were excluded from analyses. A total sample size of *n* = 8833 was used for the baseline FAVVA with healthcare costs (Part I) and *n* = 6955 for changes in the FAVVA with healthcare costs (Part II) (Figure 1).

### 2.3. Sociodemographic Characteristics and Anthropometry

The Accessibility Remoteness Index of Australia (ARIA) provided a measure of residential area, categorised as ‘major cities’, ‘inner regional’, ‘outer regional’, ‘remote’ or ‘very remote’ [16]. Financial stress was assessed by a self-reported ability to manage on their current income, collected by a single-item question and categorised as ‘easy’, ‘not too bad’, ‘difficult some of the time’, ‘difficult all of the time’ or ‘impossible’. The BMI was calculated using self-reported height and weight data and categorised as underweight (<18.5 kg/m^2^), healthy weight (18.5–24.99 kg/m^2^) or overweight/obese (≥25 kg/m^2^) [17].

### 2.4. Assessment of Dietary Intake

The Cancer Council of Victoria’s Dietary Questionnaire for Epidemiological Studies (DQES) Version 2 food frequency questionnaire (FFQ) was used to assess the dietary intake [18,19]. The DQES asks participants to report their usual consumption of 74 foods and beverages over the past 12 months on a 10-point frequency option (‘never’ up to ‘3 to 4 times/day’). Portion size photographs were used to calculate a single portion size factor (PSF) to indicate whether, on average, a person eats median-size servings (PSF = 1), more than the median (PSF > 1) or less than the median (PSF < 1) and was used to scale the reported serving sizes for vegetables, meat and casseroles.

The mean total daily F&V intake was derived from responses to individual items for fruits (11 items) and vegetables (24 items) and summed to generate a total intake of fruits, vegetables and F&V (g/day). The nutrient intakes were computed from NUTTAB 1995 by the Cancer Council of Victoria [20]. The development of DQES [21] and validation using plasma biomarkers to estimate polyunsaturated and monounsaturated fats and F&V intakes has been reported [22,23].

### 2.5. Fruit and Vegetable Variety Index (FAVVA)

The FAVVA [24] scores were derived using DQES data [18]. FAVVA captures both the frequency and variety of the F&V intake and has demonstrated moderate-to-strong positive correlations with dietary intakes of key nutrients (vitamin C, vitamin A, fibre, potassium and magnesium) and plasma concentrations of carotenoids rich in fruits and vegetables [24]. As the original FAVVA was developed using the Australian Eating Survey FFQ, the FAVVA was modified slightly in the current study to align with the data collected by the DQES FFQ. Appendix A details the scoring method for items in the modified FAVVA used in the current study, while Appendix A outlines the differences in the scoring methods for the modified and original FAVVA. For the modified FAVVA used in the current study, points were awarded incrementally based on the frequency of different types of F&V, as assessed by the DQES, such that zero points were awarded for ‘Never’, 1 point for ’Less than 1 per month’, 2 points for ‘1–3 per month’, 3 points for ‘once per week’, 4 points for ‘2–4 per week’ and 5 points for ‘5 or more per week’. The F&V typically consumed frequently (e.g., bananas and carrots) had additional response options of 5 points for 5 to 6 times per week, 6 points for ‘Once per day’ and 7 points for ‘2 or more times per day’. Additional points were awarded for the total number of different F&V servings consumed per day. The items were summed to calculate the total FAVVA scores, ranging from 0 to 185 points (a maximum of 66 points for the Fruit subscale and 119 points for the Vegetable subscale). Changes in the FAVVA were calculated by subtracting the baseline FAVVA (2001) from Survey 7 FAVVA (2013), where negative scores indicated that the FAVVA score worsened, and positive scores indicated that the FAVVA improved over time.

### 2.6. Medicare Benefit Schedule Data

Medicare is Australia’s universal healthcare coverage provided by the Australian government. Healthcare coverage under Medicare includes services that are eligible for rebate according to the Medicare Benefits Schedule (MBS). The MBS provides benefits for doctor consultations, scheduled fees for out-of-hospital services for doctors (including examinations and tests ordered by doctors), specialist consultations and service fees, many surgical and other therapeutic procedures performed by doctors, some surgical procedures performed by dentists, eye tests performed by optometrists and other allied health consultations, as well as specified items under nominated care schemes (e.g., Cleft Lip and Palate Scheme and Better Access Scheme).

The MBS variables included the number of claims made, the ‘charge’ (the total cost of the service, as charged by the provider), the ‘benefit’ (the amount paid by Medicare back to the patient) and the ‘gap’ (the difference between the charge and the benefit, i.e., the patient’s out-of-pocket or direct costs). The MBS data were provided by Medicare in 2016 for the years 2001–2015. The cumulative number of claims, charges, benefits and gap costs were calculated across the 15-year period for each woman. Zero values were assumed for women who had no records of MBS claims during 2001–2015 (*n* = 445), as data were only provided for women who had made claims. Women with the highest 1% (*n* = 128, total charges > $79,525) and lowest 1% (*n* = 445, total charges = $0) of cumulative MBS charges over the 15-year period were excluded from the analyses to avoid anomalies associated with extreme values.

### 2.7. Statistical Methods

The statistical analyses were conducted using STATA IC, Version 13 (StataCorp LP, College Station, TX, USA). FAVVA quintiles were generated using the xtile function in STATA. Descriptive statistics were calculated for sociodemographic, anthropometric and fruit and vegetable intakes for the included women at Survey 3 by FAVVA quintiles. Differences between the FAVVA quintiles were explored using chi-square analyses (categorical data) and one-way ANOVA (continuous data). The FAVVA scores were normally distributed; however, as the MBS data were highly skewed to the right, nonparametric statistics were used. Multiple linear regression modelling was performed, adjusting for the area of residence, self-reported financial stress and total energy intake at Survey 3. For the ‘Underweight’ category, regression modelling was not performed on account of the small sample size.

The Part I Baseline F&V frequency and variety with the 15-year Medicare claims/costs: Median 15-year cumulative Medicare claims, charges, benefits and gap costs were reported by the quintiles of FAVVA at Survey 3. The FAVVA quintiles were treated as a categorical predictor of each MBS variable presented by the BMI category, with FAVVA quintile 1 (lowest F&V intake) set as the reference group. In addition, the FAVVA total and Fruit and Vegetable subscales were treated as continuous predictors of the MBS variables. The Part II Change in F&V frequency and variety with the 15-year Medicare claims/costs: For each BMI category, changes in the FAVVA total and subscale scores were treated as continuous predictors of the MBS variables.

## 3. Results

Among the women with baseline dietary and MBS data (*n* = 8833), 34.6% (*n* = 2924) lived in major cities, and 62.6% (*n* = 5290) found it easy/not too bad to manage on their current income, while over half (56.2%, *n* = 4750) had a high BMI (overweight/obesity). The mean baseline FAVVA was 87.7 ± 21.2 points (maximum 185 points), the fruit intake was 216.5 ± 139.8 g/day (recommended target 300 g/day) and the vegetable intake was 176.5 ± 80.4 g/day (recommended target 375 g/day) [8].

There were no differences in the age, weight or area of residence across the FAVVA quintiles at the baseline (Table 1). The women in FAVVA quintile 1 (lowest) were more likely to find it difficult to manage on their current income and be overweight/obese compared with those in FAVVA quintile 5 (*p* < 0.05).

### 3.1. Part I: Baseline F&V Variety with 15-Year Healthcare Claims/Costs

Healthy weight women (BMI = 18.5–24.99 kg/m^2^) in the highest FAVVA quintile made six fewer claims over 15 years compared with those in quintile 1 (220 claims (95% CI 144–318) compared with 226 claims (145–358), Table 2). For all women, those in FAVVA quintiles 2–5 incurred higher gap (out-of-pocket) costs compared with those in FAVVA quintile 1.

Among healthy weight women, higher FAVVA and FAVVA Vegetable scores were associated with fewer claims and benefits (Table 3). For every 10-point-higher FAVVA, healthy weight women made, on average, 4.3 (95% confidence interval (CI) 1.7–6.8) fewer claims over 15 years. For every 10-point-higher FAVVA Vegetable score, healthy weight women made 7.1 (95% CI 3.3–10.9) fewer claims and incurred $AUD 293.0 (95% CI $8.6–57.4) less in charges. Among the women overweight/obese, a higher FAVVA was associated with higher charges and out-of-pocket (gap) costs, while a higher FAVVA Fruit score was associated with higher claims and all costs. For every 10-point-higher FAVVA, women overweight/obese incurred $187.8 (95% CI S2.4–373.2) more in charges over 15 years. For all women, higher FAVVA, Vegetable and Fruit subscales were associated with higher out-of-pocket costs.

A higher total fruit and vegetable intake (g/day) and total fruit intake were each associated with higher claims, charges and benefits for women overweight/obese only. For every 100-g-higher intake of fruit consumed per day, women overweight/obese made, on average, 3.7 (95% CI 0.9–6.4) more claims and incurred $253.6 (95% CI $52.9–454.4) more in charges (Appendix A).

### 3.2. Part II: Change in F&V Variety over Time with 15-Year Healthcare Claims/Costs

Across the sample, the mean changes in the FAVVA, FAVVA Vegetable and FAVVA Fruit scores (2001–2013) were 0.94 ± 17.1 points, 1.0 ± 11.9 points and −0.1 ± 9.0 points, respectively (*n* = 6955). The mean change in the total FAVVA was 1.7 ± 17.5 points for women with underweight BMI (*n* = 96), 1.4 ± 17.2 points for women with healthy BMI (*n* = 3007) and 0.6 ± 17.1 points for women overweight/obese (*n* = 3854).

The changes in the total FAVVA, FAVVA Vegetable and FAVVA Fruit scores were inversely associated with the cumulative total claims and charges over 15 years (Table 4). For every 10-point increase in the FAVVA over time, on average, women made 4.3 (95% CI 1.9–6.8) fewer claims and incurred $309.1 (95% CI $129.3–488.8) less in charges.

The change in the total daily grams of F&V consumed was also inversely associated with the cumulative total charges incurred over 15 years. For every 100-g-increase in the daily F&V intake over time, on average, women made 6.5 (95% CI 3.6–9.4) fewer claims and incurred $480.0 (95% CI $265.2–694.7) less in charges (Appendix A).

## 4. Discussion

In this analysis, from the 1946–1951 cohort of the Australian Longitudinal Study on Women’s Health, we found that a higher baseline F&V frequency and variety was associated with fewer cumulative 15-year healthcare claims among healthy weight women but not women overweight/obese. A higher baseline vegetable intake was associated with fewer healthcare claims for healthy weight women and increasing the F&V intake over time was associated with fewer healthcare claims and charges among all the women. These findings were largely in keeping with our expectations, given the known benefits of F&V consumption [6,7,25]. However, the positive association between fruit intake and healthcare claims and costs among women overweight or obese was unexpected and requires further investigation.

Our results indicate that increasing the variety and frequency of F&V regularly consumed over time, regardless of the baseline intake or weight status, is associated with lower healthcare costs. Improving the diet quality over time conveys a reduced risk of cardiovascular disease and all-cause mortality [26,27,28]. Despite the national dietary guidelines advocating regular F&V consumption, the current intakes remain low [3]. The Australian Dietary Guidelines recommend consuming 300 g of fruit (two servings at 150 g/serving) and 375 g of vegetables (five servings at 75 g per serving) per day [8]. The national data from 2017 to 2018 showed that fewer than one in ten Australian adults met the vegetable consumption target [29]. Given our findings, one potential strategy to address long-term healthcare costs could be the development of population health interventions that focus on increasing F&V consumption by consuming one new variety of fruit or vegetable per week. A cost-effectiveness modelling study showed that a subsidy on F&V, when combined with taxes on saturated fat, sugar, salt and sugar-sweetened beverages, could save the Australian health sector $AU 3.4 billion annually [30]. The economic modelling in Canada demonstrates that increasing the F&V consumption by one serving per day would avoid approximately $CAN 9 billion in total costs [5]. In the US, a 30% subsidy on F&V would prevent approximately 1.9 million deaths and save $US 40 billion in healthcare costs [31]. However, while public health interventions are important, multisectoral action across nutrition, agriculture and technology to increase the global supply of F&V and reduce food waste is also needed [32].

The association between lower healthcare costs and a higher vegetable variety observed among healthy weight women is consistent with the literature. We showed that a higher vegetable intake was associated with fewer healthcare services and costs over 10 years in both women of a healthy weight and women overweight [7]. The previous study assessed the vegetable intake using the vegetable subscale of the Australian Recommended Food Score (ARFS), which allocates one point for each additional type of vegetable usually consumed at least weekly. While the ARFS vegetable subscale does take into account variety, it considers frequency based only on weekly consumption and not more frequently. By comparison, the FAVVA considers the variety in addition to frequency across the full spectrum of intake, as responses from ‘never’ up to ‘3 to 4 times/day’ are used to calculate the score. Thus, a higher FAVVA score indicates both a greater frequency and variety and, hence, a greater volume (grams per day).

A higher F&V intake has been associated with a lower risk of all causes and cardiovascular mortality [1,33]. A large, 11-year cohort study (*n* = 3704) nested in the European Prospective Investigation into Cancer and Nutrition-Norfolk study found that a greater quantity of vegetables consumed was associated with a 24% reduction in the risk of developing type 2 diabetes, while a greater variety of vegetables consumed was associated with a 23% risk reduction [11]. We have previously found that higher fruit and vegetable consumptions (measured using the Fruit and Vegetable Index (FAVI)) were associated with less weight gain among young women [34]. Our study adds to this body of knowledge by demonstrating the importance of both the frequency and variety of vegetable intakes in terms of their association with future healthcare costs.

Our findings regarding the association between fruit consumption and healthcare costs are slightly at odds. While women reporting higher fruit intakes incurred fewer healthcare charges, when the data were analysed by the BMI category, this association was reversed for women overweight/obese. For those women overweight, a higher variety and frequency of fruit consumed, consistent with a higher score, could reflect either an excess overall intake or potentially be due to over-reporting of their intake or could be associated with other characteristics, such as lower physical activity levels. One cohort study reported lower healthcare charges for men with the highest fruit intake [6], although this study was not nationally representative and examined the quantity of the fruits consumed, not variety [6]. A systematic review found no association between whole, fresh fruit consumption and excess energy intake or adiposity [35]. There are likely further confounding factors, such as food insecurity which has a complex relationship with being bodyweight and healthcare expenditures in older adults [36], as well as other sociodemographic and lifestyle factors influencing the changes in diet quality over time [37].

Not all healthcare services appear in the national MBS data. The MBS scheme in Australia includes medical information relating to claims for healthcare services that are eligible for rebate under the Medicare funding scheme. Public hospital and outpatient services are not captured, as these services are funded entirely by Medicare (and are therefore not applicable for rebate). Women who do not have private health insurance are more likely to use public hospital and public outpatient services; hence, information relating to their usage of public services will not appear in the MBS data, as these public services are funded entirely by Medicare and do not appear in the MBS. Hence, our results could be affected by residual confounding due to factors and/or variables not captured, including private health insurance status and health outcomes. Our analyses only included women who had both dietary and Medicare data, possibly influencing the findings, as these women had higher health statuses and lower financial stress. We recognise that the use of self-reported methods to collect dietary data potentially overestimates the F&V intake [38]. Dietary data were only collected at the two time points in 2001 and 2013; thus, it was not possible to ascertain at what point in time between 2001 and 2013 any changes in F&V consumption may have occurred, although the dietary patterns in this cohort remained relatively stable over time [39]. Although the data were from a large cohort of mid-aged Australian women, the results may not be generalisable among the broader population. The cohort study design precluded drawing inferences regarding cause and effect. There was also likely a residual confounding due to the variables that may not have been accounted for (e.g., health status, physical activity level and smoking and alcohol intake); the current analysis adds to the current literature regarding the association between F&V intake and long-term healthcare costs.

## 5. Conclusions

A higher F&V frequency and variety is associated with a small reduction in healthcare claims for healthy weight women, although a higher fruit intake among women overweight/obese is associated with higher costs. Increasing their F&V frequency and variety over time is associated with lower healthcare costs.

## Figures and Tables

**Figure 1 nutrients-13-02829-f001:**
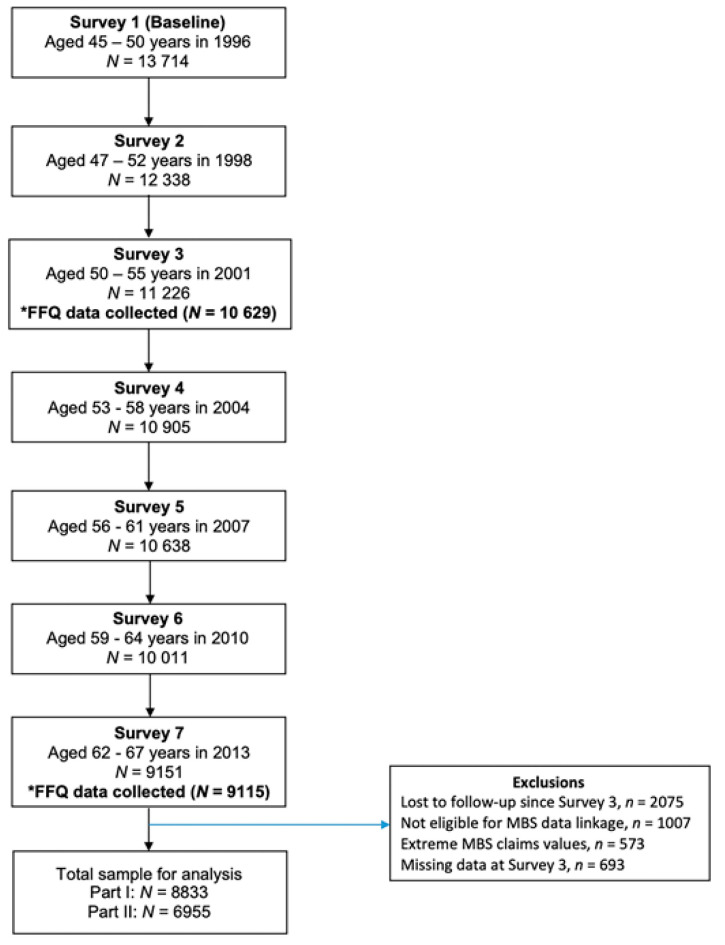
Flow chart of the participant selection using the 1946–1951 ALSWH cohort. Note: *FFQ, food frequency questionnaire; MBS, Medicare Benefits Schedule.

**Table 1 nutrients-13-02829-t001:** Characteristics of the women in the 1946–1951 ALSWH cohort according to the Fruit and Vegetable Variety (FAVVA) scores for Survey 3 (2001, *n* = 8833).

	FAVVA Quintile					
	Q1	Q2	Q3	Q4	Q5	All
*n*	1796	1840	1699	1761	1737	8833
FAVVA score ^a^	58.1 ± 10.7	77.4 ± 3.7	88.6 ± 2.9	99.1 ± 3.5	117.0 ± 10.5	87.7 ± 21.2
FAVVA range	≤70	71–83	84–93	94–105	≥106	
FAVVA Fruit ^b^	15.8 ± 6.5	22.6 ± 6.4	27.0 ± 6.0	31.3 ± 6.0	39.2 ± 7.1	27.1 ± 10.2
FAVVA Vegetable ^c^	42.3 ± 9.6	54.8 ± 6.6	61.5 ± 6.2	67.7 ± 6.2	77.9 ± 8.2	60.7 ± 14.1
Age (y)	52.5 ± 1.4	52.5 ± 1.5	52.5 ± 1.5	52.5 ± 1.5	52.5 ± 1.5	52.5 ± 1.5
Area of residence						
Major cities	35.0 (629)	33.8 (621)	36.8 (625)	33.3 (587)	33.8 (587)	34.5 (3049)
Inner regional	41.2 (740)	42.3 (779)	40.4 (687)	41.3 (728)	43.8 (761)	41.8 (3695)
Outer regional	20.7 (371)	20.2 (372)	19.5 (332)	21.8 (384)	18.7 (325)	20.2 (1784)
Remote	2.3 (41)	3.2 (59)	2.8 (48)	3.0 (52)	3.1 (53)	2.9 (253)
Very remote	0.8 (15)	0.5 (9)	0.4 (7)	0.6 (10)	0.6 (11)	0.6 (52)
Self-reported ability to manage on their current income						
Easy	14.4 (259)	19.3 (355)	18.9 (321)	19.2 (338)	21.0 (365)	18.5 (1638)
Not too bad	39.4 (708)	43.0 (791)	46.5 (790)	44.6 (786)	44.6 (775)	43.6 (3850)
Difficult some of the time	30.6 (550)	25.9 (476)	24.5 (416)	27.1 (477)	26.4 (458)	26.9 (2377)
Difficult all of the time	13.3 (239)	10.5 (194)	9.3 (158)	7.4 (131)	7.1 (123)	9.6 (845)
Impossible	2.2 (40)	1.3 (24)	0.8 (14)	1.6 (29)	0.9 (16)	1.4 (139)
Weight (kg)	71.4 ± 16.1	71.6 ± 14.9	71.0 ± 14.5	71.6 ± 14.7	71.0 ± 14.7	71.4 ± 15.0
BMI (kg/m^2^)	27.1 ± 5.9	27.0 ± 5.6	26.7 ± 5.3	26.9 ± 5.3	26.6 ± 5.2	26.9 ± 5.5
BMI category						
Underweight	1.9 (35)	1.6 (30)	1.2 (20)	0.9 (16)	1.0 (18)	1.3 (119)
Healthy weight	41.1 (738)	40.9 (753)	43.6 (740)	41.6 (732)	44.9 (780)	42.4 (3743)
Overweight/obese	57.0 (1023)	57.4 (1057)	55.3 (939)	57.5 (1013)	54.1 (939)	56.3 (4971)
Energy intake (kJ/day)	5972.7 ± 2323.3	6487.2 ± 2287.5	6625.3 ± 2237.0	6872.1 ± 2392.7	7457.2 ± 2938.7	6676.6 ± 2494.6
Fruit and vegetable intake (g/day) ^d^	268.2 ± 136.4	345.5 ± 147.5	385.7 ± 152.0	443.1 ± 163.5	528.4 ± 185.7	393.0 ± 180.6
Fruit intake (g/day) ^d^	124.9 ± 100.4	176.2 ± 119.9	211.8 ± 122.8	255.0 ± 130.3	319.4 ± 138.8	216.5 ± 139.8
Vegetable intake (g/day) ^d^	143.3 ± 81.2	169.4 ± 75.2	174.0 ± 74.5	188.1 ± 75.7	209.0 ± 80.4	176.5 ± 80.4

^a^ FAVVA maximum score = 185. ^b^ FAVVA Fruit maximum score = 66. ^c^ FAVVA Vegetable maximum score = 119. ^d^ Recommended intakes for Australians: 300 g fruit per day, 375 g vegetables per day and 675 g fruit/vegetables per day, according to the Australian Dietary Guidelines.

**Table 2 nutrients-13-02829-t002:** Median 15-year (2001–2015) cumulative Medicare claims and costs ($AU) for Australian women born in 1946–1951 by the quintile of the baseline FAVVA score (1 = lowest and 5 = highest FAVVA quintiles) and baseline BMI category (*n* = 8833) ^a^.

		Under Weight ^b^ *n* = 119		Healthy Weight *n* = 3743		Overweight/Obese *n* = 4971		ALL *n* = 8833	
BMI, Mean ± SD		17.6 ± 0.9		22.6 ± 1.6		30.4 ± 4.8		26.9 ± 5.5	
2001 FAVVA Quintile	Medicare Variable	Median	Q1, Q3	Median	Q1, Q3	Median	Q1, Q3	Median	Q1, Q3
1 (lowest)	*n*	37		748		1041		1826	
	FAVVA	62	45, 68	62	53, 70	63	53, 70	62	53, 70
	Claims (*n*) ^c^	228	147, 441	226	145, 358	266	170, 401	250	157, 387
	Charge ($) ^d^	12,580	7504, 27,566	13,262	7782, 23,063	15,169	8787, 24,673	14,261	8195, 24,246
	Benefit ($) ^e^	11,393	6798, 20,669	10,055	6066, 17,603	12,230	7223, 19,636	11,265	6620, 18,910
	Gap ($) ^f^	2190	705, 3988	2583	923, 5458	2168	759, 5631	2345	802, 5540
2	*n*	32		835		1119		1986	
	FAVVA	79	72, 85	79	73, 86	79	72, 86	79	73, 86
	Claims (*n*)	192	109, 321	219	147, 315	260	163, 381	240	156, 353
	Charge ($)	10,845	6253, 17,903	13,404	8099, 22,050	15,839	8811, 24,764	14,614	8450, 23,615
	Benefit ($)	8158	4764, 15,694	10,044	6242, 15,647	12,021	7016, 19,020	10,984	6512, 17,497
	Gap ($)	2019	878, 3858	3028 *	1320, 6087	2821	949, 6345	2886 *	1081, 6206
3	*n*	22		699		919		1640	
	FAVVA	89	85, 96	89	83, 95	89	83, 95	89	83, 95
	Claims (*n*)	226	128, 283	224 *	156, 320	269	182, 392	246	170, 363
	Charge ($)	15,045	8061, 20,111	13,553	8435, 22,154	16,770	9852, 27,132	15,290	9336, 25,061
	Benefit ($)	11,122	6818, 14,200	10,085	6605, 15,905	12,881	7772, 19,952	11,470	7132, 18,301
	Gap ($)	2850	1620, 6143	3131 *	1268, 5983	3308 *	1399, 7317	3230 *	1330, 6702
4	*n*	15		703		910		1628	
	FAVVA	100	98, 106	99	92, 106	98	91, 104	98	92, 105
	Claims (*n*)	431	202, 532	223 *	150, 326	252	167, 379	239	161, 358
	Charge ($)	22,803	15,527, 33,031	13,728	8293, 22,532	15,331	9060, 25,531	14,788	8844, 24,292
	Benefit ($)	18,016	8683, 29,663	10,327	6336, 16,445	11,754	7155, 18,994	10,859	6800, 18,019
	Gap ($)	3729	1548, 6974	3070 *	1440, 6128	3434 *	1351, 6895	3251 *	1387, 6586
5 (highest)	*n*	13		758		982		1753	
	FAVVA	114	109, 123	112	105, 120	111	104, 121	112	104, 121
	Claims (*n*)	208	176, 277	220 *	144, 318	260	167, 393	237	154, 357
	Charge ($)	13,988	11,982, 16,097	14,010	8008, 21,279	16,813	9206, 26,808	15,251	8695, 24,394
	Benefit ($)	11,357	8319, 13,101	9878	6091, 15,673	12,262	7058, 20,286	11,250	6618, 18,151
	Gap ($)	2666	686, 4740	3552	1375, 6399	3250 *	1249, 7242	3448 *	1321, 6761

Abbreviations: BMI, body mass index; FAVVA, Fruit and Vegetable Variety index; SD, standard deviation; * *p* < 0.05: linear regression modelling by BMI category with adjustment for area of residence, ability to manage on current income and total energy intake, with FAVVA Quintile 1 set as reference group. ^a^ Data presented are unadjusted medians and IQRs. ^b^ Women with ‘Underweight’ BMI were excluded from the analysis due to the small sample size. ^c^ Number of healthcare services received under the Medicare Benefits Schedule. ^d^ Total cost of services (as charged by the healthcare provider). ^e^ Amount paid back to the patient by Medicare. ^f^ Out-of-pocket costs paid by the patient.

**Table 3 nutrients-13-02829-t003:** Coefficients and 95% confidence intervals per 10-unit-higher baseline FAVVA as predictors of the 15-year (2001–2015) cumulative Medicare claims and costs for Australian women born in 1946–1951 (*n* = 8833) and within the baseline BMI category ^a^.

2001 Fruit and Vegetable Intake	Medicare Variable	Healthy Weight *n* = 3743	Overweight/Obese *n* = 4971	All *n* = 8833
**FAVVA** **Total**	Claims (*n*) ^b^	−4.3 (−6.8, −1.7) *	0.3 (−2.3, 2.9)	−1.6 (−3.4, 0.2)
	Charge ($AUD) ^c^	−132.6 (−321.6, 56.5)	187.8 (2.4, 373.2) *	47.1 (−85.7, 179.9)
	Benefit ($AUD) ^d^	−188.0 (−325.8, −50.1) *	66.9 (−69.2, 203.0)	−45.0 (−142.6, 52.5)
	Gap ($AUD) ^e^	55.4 (−10.4, 121.2)	120.9 (56.3, 185.5) *	92.2 (46.2, 138.1) *
**FAVVA** **Fruit**	Claims (*n*)	−4.9 (−10.2, 0.3)	5.6 (0.2, 10.9) *	1.4 (−2.5, 5.2)
	Charge ($AUD)	−13.9 (−409.9, 382.1)	567.4 (184.0, 950.7) *	324.8 (48.2, 601.4) *
	Benefit ($AUD)	−182.7 (−471.6, 106.2)	336.7 (55.2, 618.1) *	119.5 (−83.8, 322.8)
	Gap ($AUD)	168.8 (31.1, 306.5) *	230.7 (97.0, 364.4) *	205.3 (109.6, 301.0) *
**FAVVA** **Vegetable**	Claims (*n*)	−7.1 (−10.9, −3.3) *	−2.2 (−6.1, 1.7)	−4.4 (−7.1, −1.6) *
	Charge ($AUD)	−293.0 (−577.4, −8.6) *	122.3 (−153.6, 398.1)	−61.9 (−260.4, 136.5)
	Benefit ($AUD)	−331.4 (−538.7, −124.0) *	−26.0 (−228.4, 176.5)	−162.0 (−307.8, −16.3) *
	Gap ($AUD)	38.3 (−60.7, 137.3)	148.3 (52.1, 244.4) *	100.1 (31.4, 168.8) *

Abbreviations: BMI, body mass index; FAVVA, Fruit and Vegetable Variety index. * *p* < 0.05; linear regression modelling with adjustment for the area of residence, ability to manage on their current income and total energy intake. ^a^ Women with ‘Underweight’ BMI were excluded from the analysis due to the small sample size (*n* = 113). ^b^ The number of healthcare services received under the Medicare Benefits Schedule. ^c^ Total cost of the services (as charged by the healthcare provider). ^d^ Amount paid back to the patient by Medicare. ^e^ Out-of-pocket costs paid by the patient.

**Table 4 nutrients-13-02829-t004:** Coefficients and 95% confidence intervals (CI) per 10-unit changes in the FAVVA (2001–2013) as a predictor of the 15-year (2001–2015) cumulative Medicare claims and costs for Australian women born in 1946–1951 (*n* = 6955) and within the baseline BMI category.

Change in Intake 2001–2013	Medicare Variable	Healthy Weight *n* = 3007	Overweight/Obese *n* = 3857	All *n* = 6955
**FAVVA Total**	Claims (*n*)	−2.1 (−5.3, 1.2)	−4.9 (−8.3, −1.5) *	−4.3 (−6.8, −1.9) *
	Charge ($AUD)	−152.9 (−402.8, 97.1)	−368.1 (−623.0, −113.1) *	−309.1 (−488.8, −129.3) *
	Benefit ($AUD)	−141.3 (−322.6, 40.0)	−280.6 (−466.0, −95.3) *	−252.0 (−383.0, −121.0) *
	Gap ($AUD)	−11.6 (−99.5, 76.4)	−87.4 (−177.5, 2.6)	−57.0 (−120.0, 5.9)
**FAVVA Fruit**	Claims (*n*)	−1.9 (−8.2, 4.4)	−4.9 (−11.4, 1.6)	−4.7 (−9.3, −0.03) *
	Charge ($AUD)	−175.5 (−657.4, 306.4)	−452.8 (−933.6, 28.0)	−392.6 (−735.4, −49.7) *
	Benefit ($AUD)	−159.3 (−508.8, 190.3)	−315.3 (−664.9, 34.2)	−297.6 (−547.6, −47.7) *
	Gap ($AUD)	−16.2 (−185.8, 153.3)	−137.5 (−307.2, 32.2)	−94.9 (−215.0, 25.1)
**FAVVA Vegetables**	Claims (*n*)	−3.3 (−8.1, 1.5)	−7.3 (−12.1, −2.3) *	−6.4 (−9.9, −2.9) *
	Charge ($AUD)	−227.0 (−593.4, 139.3)	−494.9 (−860.6, −129.3) *	−421.1 (−681.3, −161.0) *
	Benefit ($AUD)	−211.6 (−477.3, 54.1)	−394.7 (−660.5, −128.8) *	−356.4 (−546.0, −166.8) *
	Gap ($AUD)	−15.5 (−144.4, 113.4)	−100.3 (−229.4, 28.9)	−64.8 (−156.0, 26.4)

Abbreviations: BMI, body mass index; FAVVA, Fruit and Vegetable Variety index. * *p* < 0.05; linear regression modelling with adjustment for the area of residence, ability to manage on their current income, total energy intake and baseline FAVVA. Note: Women with ‘Underweight’ BMI were excluded from the analysis due to the small sample size.

## Data Availability

Restrictions apply to the availability of these data.

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
