# Peer review of "Increasing Fruit and Vegetable Variety over Time Is Associated with Lower 15-Year Healthcare Costs: Results from the Australian Longitudinal Study on Women’s Health"

_nutrients, 2021, doi:10.3390/nu13082829_

Round 1

Reviewer 1 Report

This paper was aimed to investigate the association between fruit and vegetable variety and health-care cost by using one birth-cohort interviewed in two surveys within the ALSWH study. Although it's a well-written paper (but the tables' layout and format were a bit messy), from epidemiological point of view, there are some major concerns here: to assess the effect on the healthcare, some major health-related risk factors need to be considered in the analyses, such as socioeconomic status, prior-medical history, smoking etc; and given MBS data only covered women attended the private health insurance system and only a 5-year-range birth cohort be included in the study, the generalisation of current findings could be a big concern.

Reviewer 2 Report

Dear Authors, you have presented a study on vegetable and fruit intake and variety and its linkage to costs involved with health care. The findings seem to be interesting, but the word variety in this context remains unclear due to lack of information in the method section. More explainations are needed to better understand what are health care costs and what is meant with variety. There are detailed information about portions but variety is poorly explained.

Especially at the beginning of your manuscript you start with very short sentences which raise a lot of questions - see my comments in the manuscript. Please edit the text so that the readers  does not start asking questions while reading the introduction and method section.

The abstract is jumping into the context. Suggest to delete the first half sentence and to replace it with some words about the need to consume veggies and fruits.

Reviewer 3 Report

The manuscript seems in good shape generally and with some relatively minor revisions may be ready for publication.

The authors point out that women with missing or incomplete dietary data were more likely to find it difficult or impossible to make ends meet, tend to live in outer regional Australia, currently smoke, and are more likely to be sedentary, and are less likely to have a healthy BMI. This circumstance seems to call for differential weighting of data to account for variability in background variables. It would be useful for the authors to compare weighted vs. unweighted results, as a check for robustness of the results.

Was imputation attempted? Data missingness (probably missing at random rather than missing completely at random) and right-censoring could be major confounders undermining interpretations of results. Some discussion of these considerations is needed, and, if possible, the authors could attempt imputation to see if results are affected.

The linear regression modelling with adjustment for area of residence, ability to manage on current income, total energy intake, and baseline FAVVA seems appropriate. Normality apparently was satisfied, although it was not clear from the manuscript whether homoscedasticity was satisfied.

The manuscript provides a good set of recommendations for policy and practice.

The authors state that "Our findings regarding the association between fruit consumption and healthcare costs are slightly at odds. While women reporting higher fruit intake incurred fewer healthcare charges, when data were analysed by BMI category this association was reversed for women with overweight/obesity."  This statement reads like potentially a classic case of Simpson’s paradox, so it may be worthwhile for the authors to explore that possibility with an eye on the impact of confounding variables.

Overall, the manuscript reads well and seems to provide a useful contribution to the research literature.

Round 2

Reviewer 1 Report

Given the nature of the data collected, the detailed acknowledgement of limitations that related to residual confounding and generalisability provides readers enough caution and to better understand the study results and value.